# The Effects of Preserving the Diaphragm on Early and Late Outcome of Lung-Sparing Radical Surgery for Malignant Pleural Mesothelioma [note 1]

**DOI:** 10.3390/jcm11226839

**Published:** 2022-11-19

**Authors:** Michelle Lee, Luigi Ventura, Ralitsa Baranowski, Joanne Hargrave, David Waller

**Affiliations:** Department of Thoracic Surgery, Barts Thorax Centre, St Bartholomew’s Hospital, West Smithfield, London EC1A 7BE, UK

**Keywords:** pleural mesothelioma, pleurectomy decortication, diaphragm

## Abstract

Background: The accepted aim of radical surgery for malignant pleural mesothelioma (MPM) is the achievement of macroscopic complete resection (MCR) whilst reducing perioperative morbidity by preserving normal tissue. Whilst preservation of the lung by pleurectomy/decortication (PD) has become widely utilised, there remains debate regarding the management of the diaphragm. Muscle-sparing complete excision of the diaphragmatic pleura is technically challenging; thus, surgeons may proceed to extended PD with phrenectomy and possible increased morbidity or to preserve the diaphragmatic pleura at the expense of MCR with potential survival deficit. We aimed to evaluate the effects of an intentional change in protocol to diaphragm-sparing PD whilst maintaining MCR as the treatment of choice for MPM. Methods: In a series of 136 patients (111M:25F, median age 68(63–73) years) undergoing radical surgery for MPM, we identified 28 patients (22M:6F, median age 67(60–71) years) in whom MCR was achieved without phrenectomy (PD group). We compared their perioperative outcomes and survival with a historical control group of 18 patients (18M:0F, median age 69(57–78) years) in whom MCR had been achieved with phrenectomy (EPD group) but in whom there was no histological evidence of diaphragm muscle invasion and who, in retrospect, could have undergone muscle-sparing MCR if this procedure had been attempted. Results: There was no significant intergroup difference in demographics or tumour cell type; the majority of both groups were found to be epithelial (PD 85.7%, EPD 77.8%). The EPD group was found to be more locally advanced (T3 55.56%) than the PD group (T1 46.43%) (*p* = 0.03). All the following parameters were significantly reduced after PD compared to EPD: operative time (188 vs. 220 min, *p* = 0.007); duration of air leak (5 vs. 10 days, *p* = 0.001), duration of inotrope (*p* = 0.009) and post-operative hospital stay (8 vs. 13 days, *p* = 0.034). There were no significant differences (*p* = 0.123) in overall survival (OS) between the two groups, but the median survival in the PD group had not been reached at a median follow up of 33.9 (24.2–46) months. Conclusions: A surgical strategy of attempting to spare the diaphragm whilst still achieving MCR wherever possible is justified by improved perioperative outcomes without compromising OS.

## 1. Introduction

The accepted aim of radical surgery for malignant pleural mesothelioma (MPM) is the achievement of macroscopic complete resection (MCR) whilst reducing perioperative morbidity by preserving normal tissue [1]. Lung-sparing procedures are now the most common worldwide for MPM [2], and The International Association for the Study of Lung Cancer (IASLC)’s Staging and Prognostic Factors Committee (SPFC) defines pleurectomy/decortication (PD) as “parietal and visceral pleurectomy to remove all gross tumour without resection of the diaphragm or pericardium” [3]. Whilst preservation of the lung by PD has become widely utilised, there remains debate regarding the management of the diaphragm [4]. Thus, surgeons may proceed to preserve the diaphragmatic pleura at the expense of MCR with potential survival deficit or to revert to an extended PD with phrenectomy and possible increased morbidity. We have previously erred towards the latter strategy but, in this study, aimed to evaluate the effects of an intentional change in protocol to diaphragm-sparing PD whilst maintaining MCR as the treatment of choice for MPM.

## 2. Method

### 2.1. Patient Selection

In a series of 136 patients (111M:25F, median age 68 (63–73) years) undergoing radical surgery for MPM over 4 years (December 2017–August 2022) at St Bartholomew’s Hospital, we identified 28 patients (22M:6F, median age 67 (60–71) years), in whom MCR was achieved without phrenectomy (PD group). We compared their perioperative outcomes and survival with a historical control group of 18 patients (18M:0F, median age 69 (57–78) years) in whom MCR had been achieved with phrenectomy (EPD group) but in whom there was no histological evidence of diaphragm muscle invasion and who, in retrospect, could have undergone muscle-sparing MCR if this procedure had been attempted. No patients presented peritoneal complications.

### 2.2. Data Collection

Patients’ demographics and information about operative procedures or mortalities were retrieved retrospectively through the National Health Service’s Care Record Service (NHS CRS) and collected on an Excel spreadsheet. All data were anonymised to preserve the privacy of the patients.

### 2.3. Diaphragm-Sparing Macroscopic Complete Resection: Operative Technique

The initial step of the diaphragmatic pleurectomy involves separating the parietal pleura from the underlying muscle and the tendon. Typically, it is easiest to start the plane in the anterior sulcus at the edge of the pleural sheet to develop the plane. We advocate digital separation with dry gauze wrapped around the index and middle fingers, whilst gentle tension is applied to the pleural sheet, ensuring that no tearing forces are directly applied to the diaphragm. In the early stage, it is important to identify the phrenic nerve at its origin, follow its path to the diaphragm, and preserve it. Depending on the extent of the disease, it may be necessary to accept partial-thickness phrenectomy or even localised full-thickness resection of muscle to achieve the primary goal of MCR. The closure, using interrupted non-absorbable sutures, also serves to plicate the diaphragm to prevent post-operative paradoxical movement in the event of phrenic neuropraxia (Figure 1).

### 2.4. Ethics

This study is registered as a clinical audit in the department of Thoracic Surgery at Barts Health NHS Trust with an audit ID 13012. The Ministry of Ethics of the UK states that when clinical audits do not alter patients’ usual clinical management, it does not require additional patient consent or formal ethical review or approval from the NHS Research Ethics Service [5].

### 2.5. Statistical Analysis

R Core Team (R: A language and environment for statistical computing. R Foundation for Statistical Computing, Vienna, Austria 2022) was used for all statistical analyses. Distributions of quantitative variables were summarised with mean, standard deviation (SD), median quartiles and range (minimum-maximum), whereas distributions of qualitative variables were summarised with number and percentage of occurrence for each of their values. A chi-squared test (with Yates’ correction for 2 × 2 tables) was used to compare qualitative variables among groups. In case of low values in contingency tables, Fisher’s exact test was used instead. The Mann–Whitney test was used to compare quantitative variables between two groups. Kaplan–Meier curves were compared with LR (log-rank) tests. R 4.2.1 was used for computations. Significance level for all statistical tests was set to 0.05.

## 3. Results

### 3.1. Pre-Operative Patient Characteristics

In a series of 136 patients (111M:25F, median age 68 (63–73) years) undergoing radical surgery for MPM, we identified 28 patients (22M:6F, median age 67 (60–71) years) in whom MCR had been achieved without phrenectomy (PD group). We compared their perioperative outcomes with a control group of 18 patients (18M:0F, median age 69 (57–78) years) in whom MCR had been achieved with phrenectomy and in whom there was no histological evidence of diaphragm muscle invasion (EPD group). In general, there was no significant intergroup difference in pre-operative demographics. The patients were typically males in their mid-sixties with no significant cardiorespiratory comorbidity. Whilst performance status appeared to be better in the PD group and statistically significant (*p* = 0.0129), this assessment is at best subjective and is dependent on the degree of effusion control at the time of assessment (Table 1).

### 3.2. Impact of Diaphragm Preservation on Perioperative Clinical Outcome

PD was statistically significantly (*p* = 0.004) more likely to be successful on the right side, whilst most cases where PD was possible but not attempted (EPD group) occurred on the left. There was no statistically significant (*p* > 0.05) intergroup difference in tumour cell type, and the majority was found to be epithelioid type. The EPD group was found to be significantly more locally advanced (T3 55.56%) than the PD group (T1 46.43%) (*p* = 0.03). This finding was due to the higher proportion of patients with non-transmural pericardial invasion in the EPD group (*p*=0.03) (Table 2).

Diaphragm preservation was associated with shorter operative time (PD: 188 (112–271) minutes; EPD: 220 (173–385) minutes; *p* = 0.007), shorter length of stay (LOS) at hospital (PD: 8 (4–23) days; EPD: 13 (5–22) days; *p* = 0.034) and shorter duration of air leak (AL) (PD: 7 (2–20) days, EPD: 10 (4–20) days, *p* = 0.001). Interestingly, those in the PD group required a significantly shorter duration of inotropes (mean (SD), PD: 0.62 (0.63) days; EPD: 1.39 (1.2 days, *p* = 0.009)). Surgery was complicated by post-operative pneumonia (*p* = 0.739), atrial fibrillation (*p* = 0.513), and chyle leak (*p* = 1) at a similar rate in both groups. In the EPD group, three patients (16.67%) developed type 1 respiratory failure, requiring non-invasive ventilation at HDU. Reoperation for non-diaphragm-related complications was required in four patients (PD group: two patients with chylothorax; EPD group: one patient with chylothorax and one patient with a patent foramen ovale). Their perioperative course was excluded for analysis, as their LOS or duration of AL were prolonged unrelated to diaphragm. In the EPD group, one additional patient was also excluded due to prolonged LOS related to social issues (Table 3).

### 3.3. Survival

At a median follow up of 33.9 (24.2–46) months, there was no significant difference (*p* = 0.123) in the OS between the two groups; however, the median survival in the PD group has not yet been reached (Figure 2, Table 4).

## 4. Discussion

### 4.1. Summary of Results

We have successfully demonstrated that sparing the diaphragm in the conduct of pleurectomy/decortication is beneficial as it is associated with a reduction in the duration of post-operative air leak and post-operative hospital stay. Furthermore, despite the perception of technical difficulty, operative time was reduced by avoiding the use of a prosthetic patch. Importantly, by adhering to the principle of MCR, sparing the diaphragm did not compromise OS. Of interest was the difference in the laterality between the two groups, with right-sided PD predominance. It is noteworthy that the perceived difficulties of removing the pleura around the bare area of the liver did not prevent diaphragm preservation. On the left, the difficulty in achieving MCR whilst preserving enough functioning muscle around the oesophageal hiatus may explain the preference for EPD and prosthetic replacement to reduce the risk of hiatal herniation. Interestingly, the EPD group was associated with a longer duration of inotropes. We hypothesize that this may be related to a more extensive sympathectomy associated with phrenectomy, thus leading to more peripheral vasodilation and consequent hypotension.

### 4.2. Impact of Diaphragm Preservation on Perioperative Clinical Outcome

Our findings imply that preservation of the native diaphragm reduces post-operative parenchymal air leak. We postulate that some post-operative preservation of diaphragmatic motion allows more rapid adhesion between the raw diaphragmatic surface of the lower lobe and muscle and between the parietal surface and the lung. When a prosthesis is used, it will not move with respiration which may allow for a persistent basal space, preventing adhesion, thereby perpetuating the air leak.

We probably underestimate how many patients in whom we could preserve the diaphragm. Our findings show that 48 of 136 (35.3%) resected specimens showed no evidence of muscle invasion and of these 48, at least 18 (37.5%) could have avoided phrenectomy. These findings are similar to the findings of the Leicester group, who found no evidence of diaphragmatic involvement following pathological assessment of the resection specimen in 119 patients (37.9%) [4]. 

Another benefit of diaphragm sparing is the reduction in the risk of post-operative pleural empyema. Lapidot et al. reported this complication in 6.8% of patients, which was associated with prolonged air leak (PAL) and use of prosthetic mesh. Not only was post-operative stay increased but median survival was also significantly reduced [6]. We reported no cases of empyema as those in our series were less likely to have prolonged drainage due to air leak and had no prosthesis to act as a nidus of infection. Complete resection of the diaphragm can also be associated with further complications due to a prosthesis, subsequently prolonging post-operative hospital stay. Sharkey et al. reported that 6.4% of patients developed prosthesis dehiscence during the immediate post-operative period (within 30 days of operation or during their in-hospital stay) and 85% of these required reoperations for patch removal [4].

The functional benefit of diaphragm preservation remains a subject of debate. Although it is unclear “how much function is retained by a diaphragm, devoid of parietal pleura and, with a damaged or traumatised phrenic nerve”, Friedberg et al. advocate to preserve as much functional muscle as possible [7]. The only published evidence is from a small series of PD where avoidance of the diaphragm resection was associated with greater increase in FVC (+34.6 ± 17.0% versus +13.5 ± 5.4%; *p* = 0.002) and FEV1 (+29.2 ± 18.1% versus +12.1 ± 6.4%; *p* = 0.015) [8].

The dilemma facing the surgeon is that the functional benefits after lung-sparing or diaphragm-sparing surgery are likely to be related to the amount of normal tissue preserved. One may argue that the consequent improved post-operative QoL may equate to longer OS due to the avoidance of non-cancer-related complications [9]. However, the survival rates have clearly been shown to be significantly better in patients having PD and MCR. Lang-Lazdunski et al. reported comparative median survival figures of 31 months versus 16.6 months, and 5-year survival rates of 38.7% versus 19% [10]. Our immature data for survival in the PD group do also suggest favourable outcomes with a 3-year survival rate of nearly 60%.

There is another theoretical consideration which may influence the surgeon’s management of the diaphragm. It has been suggested that whilst performing maximal cytoreduction, the diaphragm should be spared to minimise intra-peritoneal seeding [9]. However, Sharkey et al. found that although abdominal disease progression was found in 23.9% of their series, it was not associated with OS nor was the degree of diaphragmatic invasion [4]. It may, therefore, theoretically be unnecessary to resect the diaphragm in all cases, and a PD could suffice.

### 4.3. Limitations

Our study is limited by the relatively small number of patients, although the control group is relevant as it represents a comparable group in terms of the disease extent. Type II error may, therefore, occur with reduced reliability of analyses. We acknowledge the potential confounding effect of excluding patients with chylothorax from analysis of post-operative LOS, but we have assumed that this complication, which occurred equally in both groups, is not related to diaphragm management.

### 4.4. Future Work

To improve validity of the study, an increased size of samples is crucial. To improve evaluation of impacts of pre-operative patient characteristics on clinical outcome, further information about pre-operative adjuvant therapy or compliance with chemotherapy between the two groups can be useful. A future randomised trial to investigate the results of diaphragm sparing would be difficult to design since it would require intra-operative assessment for inclusion which would be difficult to reverse. There is scope, however, for detailed assessment of the functional benefit of diaphragm sparing not only in muscle function but also in the consequent effect on lung function. Future study of the benefit on quality of life from diaphragm preservation and compliance with adjuvant chemotherapy or immunotherapy should be undertaken.

## 5. Conclusions

A previous publication advocated resecting the diaphragm in all cases of PD as avoidance of an R2 resection and the consequent survival deficit was held to be the main priority [4]. However, in the light of our current findings, we are impressed by the comparative survival in the face of benefits in hospital resource usage. We acknowledge the more complex surgical procedure required to separate the diaphragmatic pleura from muscle but would highlight the reduction in duration of anaesthesia from avoiding the insertion of a prosthetic neo-diaphragm. 

We conclude that, wherever possible, the muscle of the diaphragm should be preserved, including partial resection, but avoiding a prosthetic patch, as long as MCR is maintained [11]. Some may retort that, unfortunately, not all patients will present in a sufficiently early stage for it to be possible to preserve the diaphragm. It may be that these patients are best treated without resection.

## Figures and Tables

**Figure 1 jcm-11-06839-f001:**
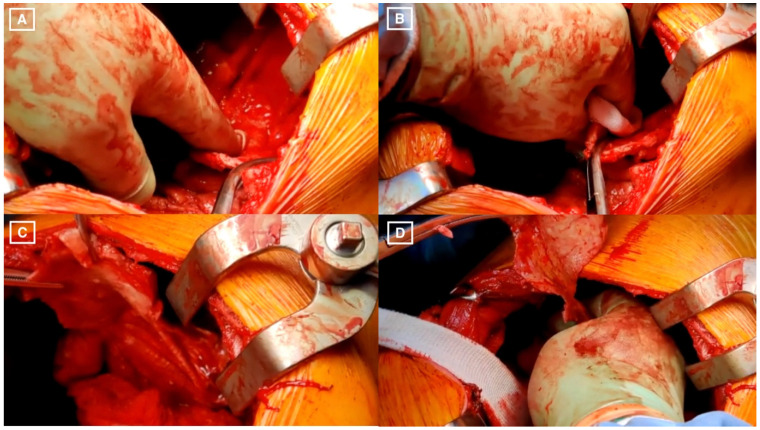
Diaphragm-Sparing Macroscopic Complete Resection: Operative Technique. (**A**) Separating the parietal pleura from the underlying muscle and the tendon; (**B**) Digital separation with dry gauze wrapped around the index and middle fingers; (**C**) Separated parietal pleura; (**D**) Further digital separation of the parietal pleura.

**Figure 2 jcm-11-06839-f002:**
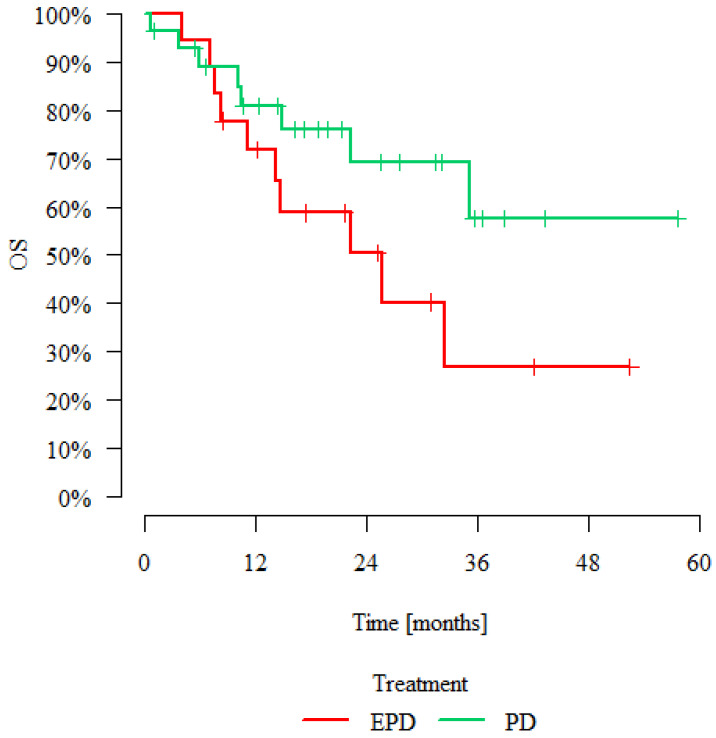
Kaplan–Meier Curve. OS indicates overall survival.

**Table 1 jcm-11-06839-t001:** Pre-Operative Patient Characteristics. EPD indicates extended pleurectomy/decortication; PD, pleurectomy/decortication; SD, standard deviation; ECOG, The Eastern Cooperative Oncology Group.

Parameter		EPD (*n* = 18)	PD (*n* = 28)	Total (*n* = 46)	*p*
Age [years]	Mean (SD)	68.78 (6.08)	65.71 (8.73)	66.91 (7.88)	*p* = 0.305
	Median (quartiles)	69 (64.75–73)	67 (60–71.25)	68.5 (62–72)	
	Range	69 (57–78)	67 (45–78)	68.5 (45–78)	
Sex	Male	18 (100.00%)	22 (78.57%)	40 (86.96%)	*p* = 0.068
	Female	0 (0.00%)	6 (21.43%)	6 (13.04%)	
Comorbidity	No	5 (27.78%)	11 (39.29%)	16 (34.78%)	*p* = 0.629
	Yes	13 (72.22%)	17 (60.71%)	30 (65.22%)	
ECOG Performance Status	ECOG 0	5 (27.78%)	19 (67.86%)	24 (52.17%)	*p* = 0.019 *
	ECOG 1	13 (72.22%)	9 (32.14%)	22 (47.83%)	
Smoking	Never	11 (61.11%)	12 (42.86%)	23 (50.00%)	*p* = 0.395
	Ex	7 (38.89%)	14 (50.00%)	21 (45.65%)	
	Current	0 (0.00%)	2 (7.14%)	2 (4.35%)	
Tumour Maximum Thickness [mm]	Mean (SD)	19.69 (17.9)	11.36 (8.24)	14.21 (12.81)	*p* = 0.073
	Median (quartiles)	15 (8–25)	9 (5–14)	9.5 (5–18)	
	Range	15 (5–64)	9 (3–32)	9.5 (3–64)	
	Unavailable	5	3	8	

* Statistically significant (*p* < 0.05).

**Table 2 jcm-11-06839-t002:** Histopathology of Malignant Pleural Mesothelioma using the eighth edition of the TNM staging system. EPD indicates extended pleurectomy/decortication; PD, pleurectomy/decortication.

Parameter		EPD (*n* = 18)	PD (*n* = 28)	Total (*n* = 46)	*p*
Laterality	Left	13 (72.22%)	7 (25.00%)	20 (43.48%)	*p* = 0.004 *
	Right	5 (27.78%)	21 (75.00%)	26 (56.52%)	
Histology	Epithelioid	14 (77.78%)	24 (85.71%)	38 (82.61%)	*p* = 0.693
	Non-Epithelioid	4 (22.22%)	4 (14.29%)	8 (17.39%)	
T staging	T1	2 (11.11%)	13 (46.43%)	15 (32.61%)	*p* = 0.03 *
	T2	5 (27.78%)	8 (28.57%)	13 (28.26%)	
	T3	10 (55.56%)	6 (21.43%)	16 (34.78%)	
	T4	1 (5.56%)	1 (3.57%)	2 (4.35%)	
N staging	N0	12 (66.67%)	20 (71.43%)	32 (69.57%)	*p* = 0.989
	N1	6 (33.33%)	8 (28.57%)	14 (30.43%)	
Regional lymph node status	Negative	10 (55.56%)	21 (75.00%)	31 (67.39%)	*p* = 0.293
	Positive	8 (44.44%)	7 (25.00%)	15 (32.61%)	

* Statistically significant (*p* < 0.05).

**Table 3 jcm-11-06839-t003:** Perioperative Clinical Outcome. EPD indicates extended pleurectomy/decortication; PD, pleurectomy/decortication; ITU, intensive therapy unit; HDU, high dependency unit; SD, standard deviation.

Parameter		EPD (*n* = 18)	PD (*n* = 28)	Total (*n* = 46)	*p*
Operation time [min]	Mean (SD)	234.12 (56.55)	190.82 (41.1)	207.18 (51.48)	*p* = 0.007 *
	Median (quartiles)	220 (201–238)	188 (164–213)	202 (173–222)	
	Range	220 (173–385)	188 (112–271)	202 (112–385)	
Intra-Operative complications	No	17 (94.44%)	28 (100.00%)	45 (97.83%)	*p* = 0.391
	Yes	1 (5.56%)	0 (0.00%)	1 (2.17%)	
Inotrope duration [days]	Mean (SD)	1.39 (1.2)	0.61 (0.63)	0.91 (0.96)	*p* = 0.009 *
	Median (quartiles)	1 (1–2)	1 (0–1)	1 (0–1)	
	Range	1 (0–5)	1 (0–2)	1 (0–5)	
	Exclusion	0	0	0	
Post-Operative ITU/HDU stay [days]	Mean (SD)	3.56 (1.31)	3.08 (1.83)	3.26 (1.65)	*p* = 0.177
	Median (quartiles)	3 (3–4.25)	3 (2–3.75)	3 (2–4)	
	Range	3 (2–6)	3 (1–8)	3 (1–8)	
	Exclusion	2	2	4	
Duration of air leak [days]	Mean (SD)	10.56 (4.62)	5.92 (3.14)	7.69 (4.36)	*p* = 0.001 *
	Median (quartiles)	10 (7–14)	5.5 (4–7)	7 (4–10.75)	
	Range	10 (4–20)	5.5 (2–13)	7 (2–20)	
	Exclusion	2	2	4	
Total length of stay [days]	Mean (SD)	12.67 (5.38)	8.92 (3.83)	10.29 (4.76)	*p* = 0.034 *
	Median (quartiles)	13 (8–16)	8 (7–9.75)	9 (7–13)	
	Range	13 (5–22)	8 (4–23)	9 (4–23)	
	Exclusion	3	2	5	
Arrythmia	No	9 (50.00%)	18 (64.29%)	27 (58.70%)	*p* = 0.513
	Yes	9 (50.00%)	10 (35.71%)	19 (41.30%)	
Chyle leak	No	17 (94.44%)	26 (92.86%)	43 (93.48%)	*p* = 1
	Yes	1 (5.56%)	2 (7.14%)	3 (6.52%)	
Respiratory failure	No	15 (83.33%)	28 (100.00%)	43 (93.48%)	*p* = 0.054
	Yes	3 (16.67%)	0 (0.00%)	3 (6.52%)	
Chest infection	No	14 (77.78%)	20 (71.43%)	34 (73.91%)	*p* = 0.739
	Yes	4 (22.22%)	8 (28.57%)	12 (26.09%)	

* Statistically significant (*p* < 0.05).

**Table 4 jcm-11-06839-t004:** Overall Survival. EPD indicates extended pleurectomy/decortication; PD, pleurectomy/decortication.

Treatment	Patients	Deaths	Overall Survival				*p*
			12 Months	24 Months	36 Months	Median [Months]	
EPD	18	10	71.79%	50.35%	26.85%	25.66	*p* = 0.123
PD	28	8	80.78%	69.12%	57.60%	>max obs.	

## Data Availability

The data used to support the findings of this study are included within the article.

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
