# Peer review of "The Effects of Preserving the Diaphragm on Early and Late Outcome of Lung-Sparing Radical Surgery for Malignant Pleural Mesothelioma [Author-notes fn1-jcm-11-06839]"

_jcm, 2022, doi:10.3390/jcm11226839_

Round 1

Reviewer 1 Report

Review of the paper titled: The Effects of Preserving The Diaphragm on Early and Late Outcome of Lung Sparing Radical Surgery For Malignant Pleural Mesothelioma by Michelle Lee and colleagues.

Comments

The paper evaluates the clinical utility of muscle-sparing complete excision of the diaphragmatic pleura compared to diaphragm resection during pleurectomy for malignant pleural mesothelioma. 

Early and late outcomes (survival) are reported among 28 patients without phrenectomy (PD group) and a control group of 18 patients, treated in the same institution. 

The paper is interesting, even if with several limitations, due to the retrospective data collection, the small sample size, the difference in T staging. 

Nevertheless, I still consider the paper interesting since it adds a different surgical opportunity, potentially able to reduce the post-op complication rate, without influencing survival. 

Following my suggestions:

-       I would avoid picture 2, since it can give to the readers an incorrect message. You should concentrate the paper on early results, and just mention long term survival whom results are not supported by data. 

-       In Discussion you should comment why the surgical time is longer in EPD than in Pd group. The reader could speculate that dissecting the diaphragmatic pleura should be time demanding…

-       In general, you should improve the “limitations” topic, giving a clear and honest message to the readers. 

Author Response

Dear Reviewer,

Thank you very much for your comments.

  1. I would avoid picture 2, since it can give to the readers an incorrect message. You should concentrate the paper on early results, and just mention long term survival whom results are not supported by data.
  • I have discussed this matter with co-authors and we felt a survival curve is important in a manuscript as it implies prediction of long term outcome.

     2.  In Discussion you should comment why the surgical time is longer in EPD than in Pd group. 

  • It is known that EPD takes longer time due to time spent for a prosthesis insertion. Therefore, it was not specifically addressed. We however concentrated on addressing MCR surgical technique in Section 2.3, which is the key of this manuscript.

     3. In general, you should improve the “limitations” topic, giving a clear and honest message to the readers.

  • Amended. Thank you.

Reviewer 2 Report

Minor revisions are needed as follows:

In Introduction section, full name of PD should be written before abbreviation. Otherwise, introduction is written informatively.

In Material and methods, the authors need to specify the period and place  in which the research was carried out. Also, it should be specified in more detail how control and experimental groups were defined. Please specify indications for phrenectomy in case of no muscle invasion of diaphragm. If there are any peritoneal complications, please specify.

In Table 2, please specify which edition of TNM was used for classification.  Please explain the difference between N staging and lymph node status.

Please specify how diaphragm was reconstructed after phrenectomy (prosthetic patch insertion?).

Were patients in both groups were subjected to any therapy regimen and which one?

Discussion is well written. References are correct.

Author Response

Dear Reviewer,

Thank you very much for your comments.

  1. In Introduction section, full name of PD should be written before abbreviation
  • Amended.

     2. In Material and methods, the authors need to specify the period and place  in which the research was carried out.

  • Amended

     3. Also, it should be specified in more detail how control and experimental groups were defined.

  • Already addressed

     4. Please specify indications for phrenectomy in case of no muscle invasion of diaphragm.

     - Already addressed in the manuscript. If no muscle invasion, PD was performed.

     5. If there are any peritoneal complications, please specify.

  • Amended

     6. In Table 2, please specify which edition of TNM was used for classification.  Please explain the difference between N staging and lymph node status.

  • Already addressed that it is 8th edition
  • Amended. Lymph node status = Regional lymph node, N staging = according to 8th edition TNM staging

     7. Please specify how diaphragm was reconstructed after phrenectomy (prosthetic patch insertion?).

  • As part of EPD, a prosthetic patch is used universally and therefore not specifically addressed in this manuscript.

     8. Were patients in both groups were subjected to any therapy regimen and which one?

  • No. As we are concentrating on the surgical techniques, we felt this may be irrelevant to the objective of the current manuscript. Therefore, this was not explored.

Kind Regards,

Michelle Lee

Reviewer 3 Report

Excellent paper from one of the best MPM centers of Europe.

Their observation on the importance of preserving the diaphragm

and limiting surgical agressivity is an important contribution in the

treatment of this malignant disease.The paper is very well written,

the analysis is correct. The message is clear and sound.

Author Response

Thank you very much.

Kind Regards,

Michelle Lee